# Heme Oxygenase Modulation Drives Ferroptosis in TNBC Cells

**DOI:** 10.3390/ijms23105709

**Published:** 2022-05-20

**Authors:** Valeria Consoli, Valeria Sorrenti, Valeria Pittalà, Khaled Greish, Agata Grazia D’Amico, Giuseppe Romeo, Sebastiano Intagliata, Loredana Salerno, Luca Vanella

**Affiliations:** 1Department of Drug and Health Science, University of Catania, 95125 Catania, Italy; valeria.consoli@phd.unict.it (V.C.); valeria.pittala@unict.it (V.P.); agata.damico@unict.it (A.G.D.); giuseppe.romeo@unict.it (G.R.); s.intagliata@unict.it (S.I.); lsalerno@unict.it (L.S.); lvanella@unict.it (L.V.); 2CERNUT-Research Centre on Nutraceuticals and Health Products, University of Catania, 95125 Catania, Italy; 3Princess Al-Jawhara Centre for Molecular Medicine, Department of Molecular Medicine, School of Medicine and Medical Sciences, Arabian Gulf University, Manama 329, Bahrain; khaledfg@agu.edu.bh

**Keywords:** HO-1, HO-2, cancer, erastin, heme, curcumin, inducers, inhibitors

## Abstract

The term ferroptosis refers to a peculiar type of programmed cell death (PCD) mainly characterized by extensive iron-dependent lipid peroxidation. Recently, ferroptosis has been suggested as a potential new strategy for the treatment of several cancers, including breast cancer (BC). In particular, among the BC subtypes, triple negative breast cancer (TNBC) is considered the most aggressive, and conventional drugs fail to provide long-term efficacy. In this context, our study’s purpose was to investigate the mechanism of ferroptosis in breast cancer cell lines and reveal the significance of heme oxygenase (HO) modulation in the process, providing new biochemical approaches. HO’s effect on BC was evaluated by MTT tests, gene silencing, Western blot analysis, and measurement of reactive oxygen species (ROS), glutathione (GSH) and lipid hydroperoxide (LOOH) levels. In order to assess HO’s implication, different approaches were exploited, using two distinct HO-1 inducers (hemin and curcumin), a well-known HO inhibitor (SnMP) and a selective HO-2 inhibitor. The data obtained showed HO’s contribution to the onset of ferroptosis; in particular, HO-1 induction seemed to accelerate the process. Moreover, our results suggest a potential role of HO-2 in erastin-induced ferroptosis. In view of the above, HO modulation in ferroptosis can offer a novel approach for breast cancer treatment.

## 1. Introduction

Ferroptosis is a recently identified type of programmed cell death (PCD), and it was first described by Dixon et al. in 2012 using RAS-selective lethal (RSL) small molecules, such as erastin and RSL3, as triggers of the process [1]. This form of PCD has characteristic features that differentiate it from other historically well-known types of cellular death such as apoptosis, autophagy, necrosis and necroptosis [2]. Ferroptosis is mainly characterized by iron accumulation, which leads to greatly increased ROS production and lipid peroxidation. This particular process involves peculiar features including mitochondrial shrinkage and dysfunction, together with the rounded morphology of the cell undergoing ferroptotic death upon erastin exposure [3]. The targets of ferroptosis inducers, for instance erastin and RSL3, are specific inhibitors of the cystine/glutamate antiporter system Xc^−^ and glutathione peroxidase 4 (GPX4), respectively. ACSL4 (acyl-CoA synthetase long-chain family member 4) dictates sensitivity to ferroptosis and, usually, its expression is upregulated in some types of cancer rather than in healthy cells, determining a potential selectivity for cancer cells [4]. Multiple lines of evidence have suggested ferroptosis’ pivotal role in tumor suppressor pathways, including p53 activation, highlighting its relevance in the field of antineoplastic therapies [5,6]. Indeed, several studies have demonstrated the high potential of ferroptosis as a novel therapeutic strategy for the treatment of different cancers, including hepatic, lung and breast cancer [7,8,9,10,11,12]. Recent studies have confirmed that female breast cancer (BC) has overtaken lung cancer as the most commonly diagnosed cancer, with an estimated 2.3 million new cases (11.7%), closely followed by lung (11.4%), colorectal (10.0%), prostate (7.3%) and stomach (5.6%) cancers. In women, BC represents the most common type of cancer diagnosed yearly, and it is the leading cause of cancer deaths [13]. BC is particularly heterogeneous, showing characteristic morphology, histology and biochemical features among the different subtypes, which, in turn, are translated into different prognosis and therapy outcomes. Triple negative breast cancer (TNBC) represents about 17% of all breast carcinomas and it is characterized by the absence of the estrogen receptor (ER), the progesterone receptor (PR) and human epidermal growth factor receptor 2 (HER2) [14]. Among other subtypes of BC, TNBC is considered the most aggressive and it has been associated with worse prognosis and outcome. Studies have shown a high recurrence potential, with the majority of deaths occurring between the third and the fifth year after initial treatment [15,16,17,18]. Due to the absence of specific molecular targets, TNBC treatment is essentially based on surgery followed by radio- and chemotherapy. However, the occurrence of relapses and metastasis spread proves the low long-term efficacy of conventional chemotherapy and facilitates the insurgence of chemo-resistance, highlighting the urgency to find alternative therapeutical strategies. Thus, ferroptosis has turned into a very captivating area of research. Among the regulators of the ferroptotic process, nuclear factor erythroid 2-related factor (Nrf2) has been gaining a great deal of attention as a multifunctional regulator of cellular redox balance and protective antioxidant response systems, including heme oxygenase (HO) enzymes. Nrf2 plays a dual role in tumor progression and its action is essentially context-dependent. Indeed, sustained Nrf2 pathway activation may provide an advantageous environment for cancer cells’ growth and survival by buffering the ROS levels that are often found to be elevated due to their abnormally rapid proliferation and fast metabolic rate [19]. Nevertheless, cancer cells show a tendency to develop a redox adaptation by promoting the activation of antioxidant and anti-apoptotic systems, enhancing cell survival and chemoresistance [20,21]. Likewise, HO, as the main target protein of Nrf2, can exert either a cytoprotective action or a detrimental one in cancer, depending on the specific cellular conditions [22,23,24]. In particular, HO-1 overexpression is generally observed in several malignant human neoplastic diseases, serving as a cytoprotective factor in the early stages of tumorigenesis, but eventually promoting malignant cells’ growth, proliferation and invasion [25,26,27,28]. On the basis of these observations, HO-1 inhibition has been widely exploited as a valid therapeutic strategy for cancer treatment [29,30,31,32,33,34,35]. However, recent findings seem to propose a different approach for pharmacological modulation of HO-1 in cancer therapy, displaying a reduction in cancerous cell proliferation following enzyme induction [36,37]. Hence, the emerging correlation between modulation of the HO system and ferroptosis can be considered as a potential new path to pursue for treating specific cancers. Taking that into consideration, in this work, we aimed to understand the impact of modulation of the HO system in ferroptosis through different approaches in order to pave the way for future studies on the mechanisms involved in the ferroptotic process and the eventual development of novel HO modulators as ferroptosis inducers and anticancer agents.

## 2. Results

### 2.1. Cellular Screening, Ferroptotic Model Validation and Implications of HO 

Based on the preliminary results (Figure 1A), the two BC cell lines MCF-7 and TNBC MDA-MB 231 showed opposites trends in terms of GPX4 and ACSL4, and low levels of HO-1. Both cell lines did not show any significant difference in their HO-2 expression levels. As shown in Figure 1B, the response to erastin (a well-known ferroptosis inducer) was significantly different in the chosen cell lines; notably, cell viability was greatly decreased after 48 h of treatment in MDA-MB 231 compared with MCF-7. In order to validate the ferroptotic model, we treated cells with known ferroptosis inhibitors: deferoxamine (DFO), ferrostatin-1 (Fer-1) and Trolox. As shown in Figure 1C, all the inhibitors were able to reverse erastin’s effect on cell viability. We further examined HO-1′s involvement in erastin-induced ferroptosis by measuring the HO-1 protein levels and enzymatic activity after treatment (Figure 1D). Analysis of the data revealed a significant increase in HO-1 levels after 36 h of treatment, as well as an increase in bilirubin formation after 48 h (data not shown). In agreement with Kwon et al. [38], erastin-mediated ferroptosis was found to increase HO-1 at both the enzymatic and transcriptional levels. Thus, subsequent treatment with an HO inhibitor could have been useful for assessing HO’s enzymatic activity contribution. Cells were treated with non-toxic concentrations of a well-known HO inhibitor (SnMP) and its enzymatic substrate (hemin). Co-treatment with SnMP and erastin surprisingly reversed erastin’s cytotoxic effect (Figure 1E), while co-treatment with hemin markedly potentiated it (Figure 1F). On the basis of these results, HO-1 siRNA was used. As shown in Figure 1G, although erastin’s cytotoxic effect on HO-1-silenced cells was slightly but significantly reduced, co-treatment with hemin was considerably less effective compared with HO-1-expressing cells.

### 2.2. Effects of Hemin and Erastin on Ferroptosis-Associated Oxidative Stress

Measurement of ROS at two different time points displayed an increase after 24 h of erastin treatment, while co-treatment with hemin showed a peak after only 8 h (Figure 2A). The increase in ROS was associated with a significant decrease in GSH cellular content, as shown in Figure 2B, especially for the combination of hemin and erastin. Concurrently, Fe^2+^ and lipid hydroperoxide (LOOH) levels were elevated after 24 h of treatment; in particular, Fe^2+^ levels were similar for the erastin and co-treatment groups, while LOOH levels were significantly increased only after the combination treatment (Figure 2C,D). Hence, inducing HO-1 may alter redox homeostasis, leading to lipid peroxidation.

### 2.3. Effects of Curcumin-Mediated HO-1 Induction

As shown in Figure 3A,B, curcumin (5–50 µM) reduced cell viability in a dose-dependent manner and, as assessed by an ELISA assay, concurrently increased HO-1 protein expression levels. Based on these results, a curcumin concentration of 30 µM was used to perform the following experiments. As well as erastin, curcumin was able to induce intracellular ROS, decrease GSH content and increase both Fe^2+^ and LOOH levels (Figure 3C–F).

### 2.4. Effect of Ferroptosis Inducers on Mitochondrial Dysfunction and Lipid ROS Accumulation

In order to confirm the activation of ferroptosis by the tested compounds, we used erastin, hemin and curcumin to perform JC-1 assay and measure lipid ROS accumulation with BODIPY 665/676. As shown in Figure 4A,B, both the erastin/hemin combination and curcumin markedly decreased mitochondrial membrane potential, as shown by the reduction in JC-1 dimer formation (red fluorescence) and an increase in green monomer fluorescence. Conversely, BODIPY fluorescence displayed a significant increase, mainly for the erastin/hemin combination and the curcumin treatment (Figure 4C).

### 2.5. Effects of Ferroptosis Inducers on GPX4, FHC and HO-2 Protein Expression

Western blot analysis was performed for GPX4, FHC (ferritin heavy chain) and HO-2 proteins (Figure 5A). As previously demonstrated [39,40,41], erastin significantly reduced GPX4 expression after 24 h of treatment, as well as the erastin/hemin combination, which did not further enhance the effects of erastin. Hemin treatment caused a significant increase in GPX4 expression levels, as also reported by Jin et al. [42].

Regarding the curcumin treatment, the 30 µM concentration was not able to reduce GPX4 levels, while 50 µM was as effective as erastin (Figure 5B).

In our experimental conditions, erastin treatment for 24 h was not able to reduce FHC levels; instead, the erastin/hemin combination and curcumin (50 µM) treatment produced a significant decrease (Figure 5C).

Surprisingly, we observed a significant reduction in HO-2 protein levels in the erastin and erastin/hemin combination groups of 23% and 30%, respectively (Figure 5D), suggesting the possible implication of HO-2 in erastin-induced ferroptosis.

### 2.6. Investigating HO-2′s Involvement in Erastin-Induced Ferroptosis

Subsequently, we tested an HO-2-selective inhibitor, LS 2/26 [43], to investigate HO-2′s role in the ferroptotic process. Prior to assessing the effect of the HO-2 inhibitor in erastin-induced ferroptosis, we determined its cytotoxicity at 5 and 10 µM, then co-treatment with erastin was examined. After 48 h, an MTT assay was performed to assess cell viability, and LS 2/26 showed a synergistic effect when combined with erastin, potentiating its cytotoxicity, as observed in Figure 5E.

## 3. Discussion

HO-1 overexpression has been demonstrated as a mechanism of resistance and malignancy progression in cancer cells; however, it has been demonstrated that low levels of HO-1 are associated with an increased risk of metastasis in oral and tongue squamous cell carcinoma, together with a significant presence of undifferentiated cells [44]. Additionally, data analysis by Han et al. showed that low expression levels of HO-1 in renal carcinoma predicted poor prognosis, which might be improved by activating ferroptosis through the induction of HO-1 [45]. Furthermore, overexpression of HO-1 in prostate cancer cells resulted in markedly reduced cell proliferation and migration [46]. Thus, the influence of HO-1 on cancer proliferation and metastasis is not univocal and may depend on the type of cancer.

The principal aim of this work was to investigate the implications of the HO system in the onset of ferroptosis. In particular, we firstly hypothesized that HO induction could accelerate ferroptosis in cancer cells expressing low levels of the inducible isoform (HO-1). Next, in order to find the most sensitive cancer cells to ferroptosis, we screened five cell lines with different tissue origins (A549 and NCI-H292 are lung cancer cell lines, MCF-7 and MDA-MB 231 are breast cancer cell lines, and DU145 is a prostate cancer cell line), measuring the protein expression levels of the most relevant ferroptotic biomarkers such as GPX4 and ACSL4, together with HO-1 and HO-2 (Figure 1A). On the basis of the results obtained from the cell viability assay, we chose to perform further experiments only on the MDA-MB 231 cell line, which turned out to be the most sensitive. We further examined HO-1′s involvement in erastin-induced ferroptosis by measuring HO-1 protein levels and enzymatic activity after treatment. In order to investigate the role of enzymatic modulation of HO in the ferroptotic process, we treated cells with non-toxic concentrations of a well-known HO inhibitor (SnMP) and its enzymatic substrate (hemin). SnMP inhibited heme degradation, leading to a reduction in HO-derived labile iron, which was initially increased by erastin. Furthermore, hemin, as an HO substrate, is known to induce both HO-1 expression and enhance its enzymatic activity [47,48], resulting in an increase of iron, which under normal conditions, is not able to induce any toxicity; nevertheless, hemin accelerates ferroptotic cell death in the presence of erastin. Hemin and ferric ammonium citrate were found to promote erastin-induced ferroptosis; however, this effect was not observed after biliverdin or bilirubin treatment [49]. As reported by Naveen Kumar [50], hemin represents a pro-oxidant molecule that can induce ferroptosis both via proteosome and inflammasome activation and Fe2+ accumulation via HO-1 induction. Additionally, low levels of hemin can induce BACH-1 degradation and Nrf2 transcriptional activity in response to augmented oxidative stress levels. Thus, it seems evident that hemin’s contribution to iron release, mostly due to HO-1-mediated heme degradation, can synergistically enhance and accelerate erastin’s cytotoxic effect, leading to a further increase in ROS and a subsequently altered cellular redox balance, which determines the pro-oxidant cascade initiation that triggers ferroptotic cell death.

To better understand and elucidate HO-1′s involvement in this process, we decided to use HO-1 siRNA. The obtained results indicated a potential additional mechanism of action for the effect of the erastin/hemin combination. Indeed, under our experimental conditions, the hemin treatment did not affect cell viability; nevertheless, it is known that hemin is able to induce ROS even at low and/or non-toxic concentrations [51]. Seiwert and colleagues have shown that HO-1 silencing increased ROS production and enhanced hemin cytotoxicity [52]. These data are in agreement with our results showing that combination treatment of erastin and hemin maintains a cytotoxic effect in HO-1-silenced cells besides HO-1 induction. Taken together, these results highlight the role of HO as one of the many factors that are able to sustain ferroptosis. Thus, it is worth remarking on the difference between the two pharmacological strategies used by us and other research groups [29,37,53], exploiting either the induction or inhibition of the HO system in order to understand their effectiveness in different cellular and animal models. Low basal HO-1 levels appear to be critical to predict cells’ sensitivity to ferroptosis, suggesting its potential targeting as a novel therapeutic approach for BC.

The antineoplastic effect of HO-1 induction is reflected by an increase om oxidative stress, which drives the progression of ferroptosis. As shown in Figure 2, the erastin and hemin co-treatment led to markedly increased ROS production, GSH depletion and lipid peroxidation concurrently with Fe^2+^ accumulation, reflecting the onset of the ferroptotic process.

Based on data from the literature [54,55,56,57], we decided to use curcumin as a naturally derived HO-1 inducer to explore the possible mechanism of HO-1 from a different perspective, which excluded the involvement of the exogenous substrate. Thus, curcumin can be considered as an inducer of ferroptosis in MDA-MB 231, in agreement with the results obtained by Cao et al. [58]. The effects of the two ferroptosis inducers on mitochondrial dysfunction and lipid ROS accumulation were assessed fluorometrically by measuring lipid ROS accumulation with BODIPY 665/676 and mitochondrial dysfunction via a JC-1 assay. The results confirmed that both curcumin and the erastin/hemin combination are able to induce ferroptosis. Even though erastin showed a low efficacy in these experiments, this can be explained by the time point chosen (8 h) for performing the experiments, displaying the higher potency of the erastin/hemin combination and curcumin alone for lipid ROS accumulation.

Western blot analysis was performed to assess the ferroptosis inducers’ effects on MDA-MB 231 protein expression levels; in particular, GPX4, FHC and HO-2 were analyzed. GPX4 is considered one of the principal ferroptotic markers and plays a pivotal role in the onset of ferroptosis, as its inhibition triggers lipid peroxidation, leading to cellular death. The results obtained might be a consequence of the cellular compensatory effect resulting from hemin-induced Nrf2 activation, as demonstrated by several studies [59,60,61]. Although Nrf2 stabilization occurs following hemin treatment, HO-1 over-activation becomes cytotoxic due to an excessive increase in iron beyond the ferritin buffering capacity. Labile iron is largely considered to be one of the main ferroptotic features, acting as an intracellular pro-oxidant that is able to trigger the Fenton reaction and, consequently, lipid peroxidation [6,62,63]. Among the several factors that influence iron metabolism, ferritin has been studied to understand the release/accumulation mechanism of labile iron. Ferritin is a cytosolic iron storage protein consisting of two subunits, namely ferritin heavy chain (FHC) and ferritin light chain (FTL). FHC has ferroxidase activity and is able to convert Fe^2+^ to Fe^3+^ in order to ensure iron’s entrance to its core [64]. Recently, new evidence has shown the significance of ferritin degradation in erastin-induced ferroptosis, highlighting the involvement of a novel identified autophagic process known as ferritinophagy, which was found to promote ferroptosis through extensive ferritin iron release [65,66]. Ultimately, we wanted to investigate the potential role of HO-2, the constitutive enzymatic isoform, whose involvement in cancer and ferroptosis is still poorly understood. The data obtained suggest a plausible implication of HO-2 in erastin’s mechanisms of action; once again, the effects of the hemin/erastin combination were more accentuated. Although few studies have focused their attention on HO-2′s role in cancer, our results seem to be in accordance with previous in vitro and in vivo evidence on HO-2 deletion, which is associated with an increase in superoxide and elevated oxidative stress levels [67,68,69].

In conclusion, ferroptosis has emerged as promising alternative to conventional cancer treatments and for its potential use in overcoming multidrug resistance, which often occurs following apoptosis-based gold standard protocols. To date, however, a full understanding of ferroptosis mechanisms is still lacking, and much more is needed to actually be able to modulate and exploit it as alternative strategy for cancer treatment. Our study focused on the modulation of one of the possible factors implicated in ferroptosis, the heme oxygenase system. Recent studies have pointed out HO’s possible involvement in this process, as it is the principal enzyme involved in heme catabolism and its activity is strictly related to intracellular iron release. Our results confirmed the initial hypothesis that cells with low HO expression can be more sensitive to ferroptosis, and that enzyme induction in this context can be useful to accelerate it. Indeed, HO can be considered as a modulator factor in ferroptosis, also representing a marker of cells’ responsiveness to this type of PCD. Several natural compounds have found to possess strong potential for clinical use as promoters or inhibitors of ferroptosis [70], but few studies have focused their attention on the involvement of the HO system. Although a substantial amount of data on ferroptosis inhibitors already exist [71], for inducers such as curcumin, more extensive studies need to be performed. Furthermore, it was interesting to notice that HO-2, the constitutive isoform of the enzyme whose role in cancer is still unclear, seems to be involved in the erastin-triggered process. Of course, these findings may lay the foundations for future studies to deepen our understanding of the role of both HO-1 and HO-2 in ferroptosis, making HO modulation a possible novel therapeutic strategy.

## 4. Materials and Methods

### 4.1. Cell Culture and Viability Assay

Experiments were conducted on the human breast adenocarcinoma cell lines MCF-7 and MDA-MB 231 (ATCC, Rockville, MD, USA). Cells were cultured in Dulbecco’s modified Eagle’s medium (DMEM), high glucose (HG) supplemented with 10% FBS and 1% penicillin–streptomycin, and maintained at 37 °C and 5% CO_2_. Both cell lines were treated with erastin (Sigma, St. Louis, MO, USA) at 1, 5, 10, and 50 µM for 48 h hours, then only MDA-MB 231 cells were treated with erastin (1, 5, 10, 50 µM) with deferoxamine (DFO 100 µM), ferrostatin-1 (1 µM) and Trolox (100 µM); hemin (5 µM), tin mesoporphyrin (SnMP, 10 µM) or curcumin (5, 10, 20, 30, 50 µM) for 48 h. Cells were also treated with a selective HO-2 inhibitor ((2-[[4-(1 H-imidazol-1-yl)butyl]thio]-5-chlorobenzothiazole)) (5, 10 µM), as previously reported by Salerno et al. [43]. In order to evaluate cell viability, cells were seeded into 96-well plates at a density of 7.0 × 10^3^ cells/well in 100 µL of the culture medium. After 24 h, treatments were administered using DMEM supplemented with 1% FBS and after 48 h, 100 µL of 0.25 mg/mL 3-(4,5-dimethylthiazol-2-yl)-2,5-diphenyltetrazolium bromide (MTT) (ACROS Organics, Antwerp, Belgium) solution was added to each well, and the cells were incubated for 2 h at 37 °C and 5% CO_2_. After incubation, the supernatant was removed, and 100 µL of DMSO was added to each well to dissolve the formazan salts produced by mitochondria. The amount of formazan was proportional to the number of viable cells in the sample. Ultimately, absorbance (OD) was measured in a microplate reader (Biotek Synergy-HT, Winooski, VT, USA) at λ = 570 nm. Eight replicate wells were used for each group, and at least two separate experiments were performed.

### 4.2. Cell Transfection

MDA-MB 231 cells were seeded 24 h prior to transfection using an Opti-MEM medium supplemented with 10% FBS and without antibiotics. Upon reaching 70–80% confluence, the cells were transfected with Lipofectamine 2000 following the manufacturer’s instructions. siRNA against HMOX-1 (5′-CCUCAAAUGCAGUAUUUUUtt-3′, Ambion by Life Technologies, Carlsbad, CA, USA) was resuspended in RNAse-DNAse free H_2_O and diluted in Opti-MEM medium without antibiotics, then the siRNA–Lipofectamine complex (ratio 1:1) was obtained after mixing and incubation for 5 min. After incubation, the culture medium was replaced and the siRNA–Lipofectamine complex was maintained for 6 h, then the cells were treated for 48 h and an MTT assay was performed. Validation of gene silencing was obtained by performing qRT-PCR.

### 4.3. Western Blot Analysis

A preliminary Western blot analysis was conducted to investigate the basal levels of HO-1, HO-2 and the ferroptosis markers GPX4 and ACSL4 in the A549 (ATCC, CCL-185), NCI-H292 (ATCC, CRL-1848), MCF-7, MDA-MB 231 and DU145 (ATCC, HTB-81) cell lines in order to choose the most sensitive to ferroptosis. Subsequently HO-2, GPX4 and FHC levels were evaluated after treatment with erastin, hemin, an erastin/hemin combination and curcumin. Samples were processed as previously described by Sorrenti et al., [37]. Membranes were incubated overnight with HO-1 (GTX101147, diluted 1:1000, GeneTex, Irvine, CA, USA), HO-2 (SPA897, diluted 1:2000, Enzo Life Sciences, Farmingdale, NY, USA), GPX4 (ab125066, diluted 1:1000, Abcam, Cambridge, UK), ACSL4 (PA5–27137, diluted 1:1000, Thermo Fisher Scientific, Rodano, MI, Italy), FHC (ab81444, diluted 1:1000) and β-actin (GTX109639, diluted 1:7000, GeneTex) primary antibodies. Goat anti-rabbit secondary antibody was used to detect blots (diluted 1:7000). Blots were scanned, and densitometric analysis was performed with the Odyssey Infrared Imaging System (LI-COR, Milan, Italy). Values were normalized to β-actin.

### 4.4. Measurement of HO Enzymatic Activity

HO’s enzymatic activity was measured in cell lysates as the difference in absorbance between 464 and 530 nm of the bilirubin produced. The reaction mixtures consisted of 20 mM Tris-HCl at pH 7.4 (2 mg/mL), the cell lysate, 0.5–2 mg/mL biliverdin reductase, 1 mM NADPH, 2 mM glucose 6-phosphate (G6P), 1 U G6P dehydrogenase and 25 µM hemin. Incubation was carried out in a circulating water bath in the dark for 1 h at 37 °C. The reaction was stopped by adding chloroform. After recovering the chloroform phase, the amount of bilirubin that was formed was measured with a double-beam spectrophotometer at OD 464–530 nm (extinction coefficient: 40 mM/cm^−1^ for bilirubin). One unit of the enzyme was defined as the amount of enzyme catalyzing the formation of 1 nmol of bilirubin/mg protein/h.

### 4.5. Determination of HO-1 Levels (ELISA)

HO-1 levels were assessed using Simple Step ELISA (ab207621, Abcam, Cambridge, UK) according to the manufacturer’s instructions. A microplate reader was used to measure the absorbance (OD) at λ = 450 nm. All samples were measured in triplicate and the results are expressed as a percentage of the control.

### 4.6. Measurement of Intracellular Fe^2+^ Content

The intracellular content of ferrous iron was determined in cells treated for 18 h using the colorimetric iron assay kit from Abcam (ab83366, Abcam, Cambridge, UK) following the manufacturer’s instructions. The assay was performed in triplicate for every sample. The results are expressed in picomoles/µg of proteins.

### 4.7. Measurement of Lipid Peroxidation

Levels of LOOH were evaluated through the oxidation of Fe^2+^ to Fe^3+^ in the presence of xylenol orange (12709580). The assay mixture contained 200 μg of the sample (total cell lysate), 100 μM xylenol orange, 250 μM ammonium ferrous sulphate, 90% ethanol, 4 mM butylated hydroxytoluene and 25 mM H_2_SO_4_. Samples were incubated at room temperature for 30 min, and the absorbance was finally measured at λ = 560 nm using a microplate reader (Biotek Synergy-HT, Winooski, VT, USA). Calibration was performed using hydrogen peroxide (0.2–20 μM). The results from at least two experiments were expressed as a percentage of the control. In order to assess the lipid peroxidation levels, cells were maintained for 30 min with fluorescent staining with BODIPY 665/676 (5 μM), then washed with PBS twice. Fluorescence was measured with a VariosKan plate reader (Thermo Fisher Scientific, Waltham, MA, USA), and the results were expressed as the fluorescence intensity (AU).

### 4.8. Measurement of Mitochondrial Membrane Potential

Following the treatments, cells were incubated with a 3 μM JC-1 staining solution (T3168, Invitrogen, Waltham, MA, USA) at 37 °C for 20 min and then washed with PBS in order to visualize the fluorescence with a fluorescence microscope (EVOS Fl AMG). The JC-1 probe aggregates to form a polymer in the mitochondrial matrix of healthy cells, producing a strong red fluorescence (Ex  =  585 nm, Em  =  590 nm). Otherwise, dysfunctional mitochondria present JC-1 monomers, resulting in the emission of a green fluorescent signal (Ex  =  514 nm, Em  =  529 nm). The results are expressed as the ratio of red/green fluorescence.

### 4.9. Determination of Thiol Groups 

The concentration of non-protein thiol groups (RSH), reflecting about 90% of the GSH cellular content, was measured in total cell lysates. After 18 h of treatment, RSH levels were evaluated by a spectrophotometric assay based on the reaction of thiol groups with 2,2-dithio-bis-nitrobenzoic acid (DTNB). A DTNB solution and the samples were mixed and incubated at room temperature for 20 min in the dark until the noticeable appearance of a yellow color. After incubation, samples were centrifuged at 3000 rpm for 10 min. The supernatant was collected and set in a black 96-well plate for measurement of the absorbance in a microplate reader (Biotek Synergy-HT, Winooski, VT, USA) at λ = 412 nm. Experiments were conducted in quadruplicate. The results are expressed in pmoles/µL.

### 4.10. Measurement of ROS Levels

Levels of reactive oxygen species (ROS) were determined using the fluorescent probe 2′,7′-dichlorofluorescein diacetate (DCFH-DA). Cells were washed with a 0.1% Triton solution to enhance cellular probe permeation, then 100 µL of a DCFH-DA working solution (200 µM) was added to each well and incubated at 37 °C for 30 min. After incubation, fluorescence was measured in a microplate reader (excitation, λ = 488 nm; emission, λ = 525 nm). Eight replicate wells were used for each group. The results are expressed as fluorescence intensity (AU)/proteins (mg/mL).

### 4.11. Statistical Analysis

At least three independent experiments were performed for each analysis. The statistical significance (*p* < 0.05) of the differences between the experimental groups was determined by Fisher’s method for analyses of multiple comparisons. For comparison between treatment groups, the null hypothesis was tested by either a single-factor analysis of variance (ANOVA) for multiple groups or an unpaired t-test for two groups, and the data are presented as means ± SEM.

## Figures and Tables

**Figure 1 ijms-23-05709-f001:**
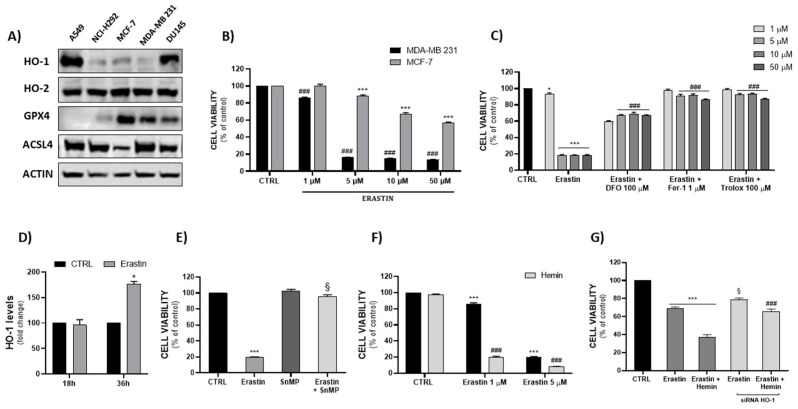
(**A**) Basal protein expression screening of HO-1, HO-2, GPX4 and ACSL4 in A549, NCI-H292, MCF-7, MDA-MB 231 and DU145 cell lines. (**B**) Evaluation of erastin’s cytotoxicity in breast cancer cell lines at 48 h (^###^ *p* < 0.0005 vs. CTRL MDA-MB 231; *** *p* < 0.0005 vs. CTRL MCF-7). (**C**) Ferroptosis model validation: erastin’s effect was reversed by DFO (100 µM), Fer-1 (1 µM) and Trolox (100 µM) at 48 h (* *p* < 0.05, *** *p* < 0.0005 vs. CTRL; ^###^ *p* < 0.0005 vs. erastin). (**D**) Measurement of HO-1 levels after treatment with erastin (5 µM) (* *p* < 0.05 vs. CTRL). (**E**,**F**) Assessment of HO-1′s modulation effect on erastin-induced cell death (*** *p* < 0.0005 vs. CTRL; § *p* < 0.0005 vs. erastin; ^###^ *p* < 0.0005 vs. erastin). (**G**) Effect of HO-1 silencing on cell viability after treatment with erastin and a erastin/hemin combination at 48 h (*** *p* < 0.0005 vs. CTRL; § *p* < 0.0005 vs. erastin; ^###^ *p* < 0.0005 vs. erastin/hemin). The results are expressed as means ± SEM.

**Figure 2 ijms-23-05709-f002:**
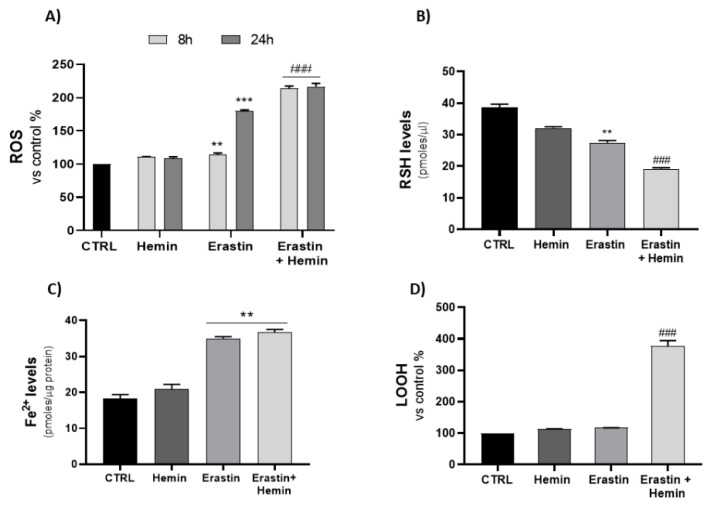
(**A**) Evaluation of the effects of hemin, erastin and an erastin/hemin combination treatments on ROS levels (** *p* < 0.005, *** *p* < 0.0005 vs. CTRL; ^###^ *p* < 0.0005 vs. erastin). (**B**) Measurement of non-protein thiol group (RSH) concentrations after 24 h of treatment (** *p* < 0.005 vs. CTRL; ^###^ *p* < 0.0005 vs. erastin). (**C**,**D**) Evaluation of Fe^2+^ and LOOH levels after 24 h of treatment (** *p* < 0.005 vs. CTRL; ^###^
*p* < 0.0005 vs. erastin). The results are expressed as means ± SEM.

**Figure 3 ijms-23-05709-f003:**
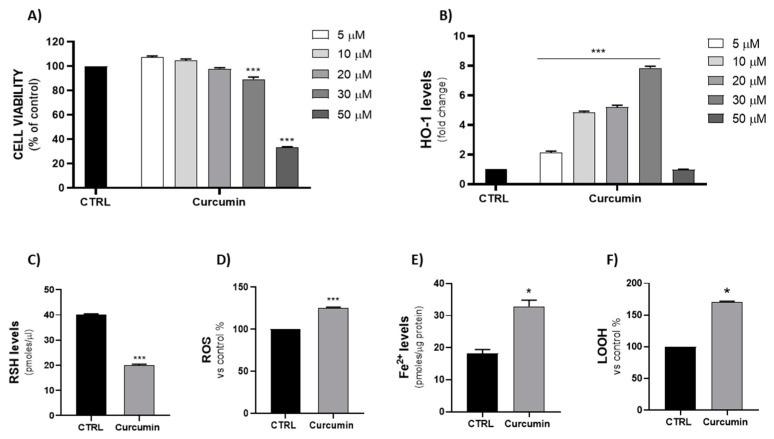
(**A**,**B**) Evaluation of curcumin (5, 10, 20, 30, 50 µM) treatment on MDA-MB 231 cells’ viability and HO-1 expression levels (*** *p* < 0.0005 vs. CTRL). (**C**–**F**) Effect of curcumin (30 µM) on RSH, ROS, Fe^2+^ and LOOH levels (* *p* < 0.05, *** *p* < 0.0005 vs. CTRL). The results are expressed as means ± SEM.

**Figure 4 ijms-23-05709-f004:**
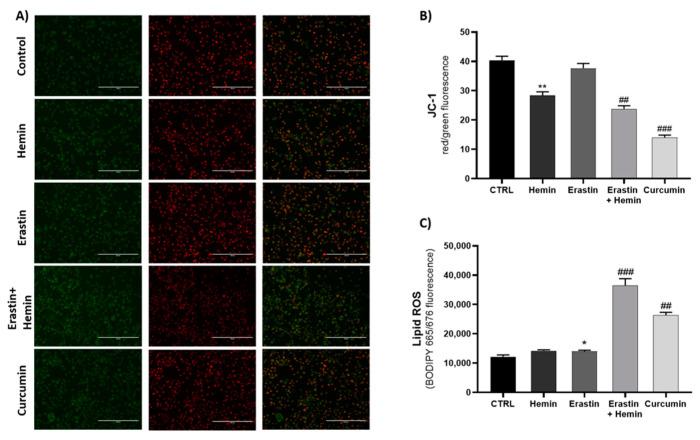
(**A**) Fluorescence images of JC-1 staining after 8 h of treatment. (**B**,**C**) Evaluation of treatment effects on mitochondrial membrane potential and lipid ROS accumulation (* *p* < 0.05, ** *p* < 0.005, vs. CTRL; ^##^ *p* < 0.005, ^###^ *p* < 0.0005 vs. erastin). The results are expressed as means ± SEM.

**Figure 5 ijms-23-05709-f005:**
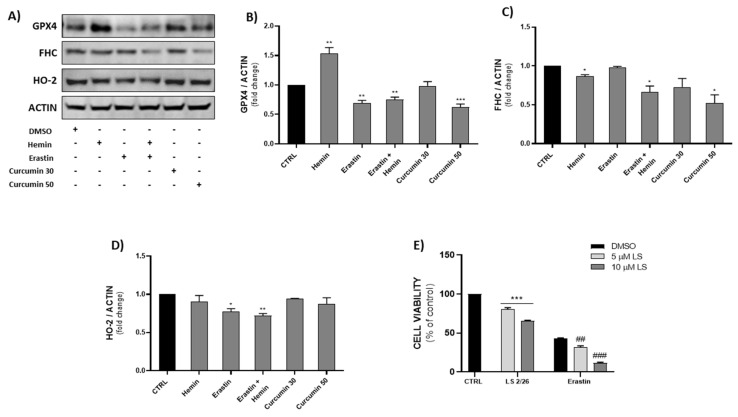
(**A**–**D**) Effects of hemin (5 µM), erastin (5 µM), the erastin/hemin combination and curcumin (30, 50 µM) on GPX4, FHC and HO-2 protein expression levels (* *p* < 0.05, ** *p* < 0.005, *** *p* < 0.0005 vs. CTRL). (**E**) Assessment of the effects of the HO-2 selective inhibitor LS 2/26 on cell viability (*** *p* < 0.0005 vs. CTRL; ^##^ *p* < 0.005, ^###^ *p* < 0.0005 vs. erastin). The results are expressed as means ± SEM.

## Data Availability

The data presented in this study are available in this paper.

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
