# Peer review of "Heme Oxygenase Modulation Drives Ferroptosis in TNBC Cells"

_ijms, 2022, doi:10.3390/ijms23105709_

Round 1

Reviewer 1 Report

Ferroptosis is a recently identified type of cell death that is morphologically genetically and mechanistically distinct uniqueness from regulated cell death, including apoptosis, necroptosis, and autophagy.  In this Manuscript, Valeria Consoli et al., investigated mechanism of ferroptosis in different types of breast cancer cells and reveal the importance of heme-oxygenase modulation. Here, author have screened initially basal protein expression , Evaluation of erastin cytotoxicity in breast cancer cells, Ferroptosis model validation, measurement of Ho1 levels after treatment with erastin and Effect of Ho1 silencing on cell viability after treatment with erastin and erastin/hemin combination and then further investigated their effect and different combination studies  such as effect of hemin and erastin on ferroptosis-associated oxidative stress, effect of curcumin-mediated ho1 induction, effect of ferroptosis inducers and ho2 protein expression and effect of ferroptosis inducers on mitochondrial dysfunction and lipid ROS accumulation. In general, well designed experiment, presented well to exhibit ho modulation is a possible novel therapeutic strategy.

Based on the results, minor questions to be addressed.

a. ROS increase was associated with a significant decrease in GSH cellular content. did you tested that  increased ROS prodution cause/result of  DNA damage.

b. Effect of ferroptosis inducers on mitochondrial dysfunction, did you analyzed mitochondrial function (cell mito stress test) using seahorse exptt. This assay provides insight into the cause of mitochondrial dysfunction  and an in-depth understanding of metabolic pathways, signals.

Fig.4a pixel quality need to be improved.

original image/blot need to be labeled.

Author Response

We appreciate reviewers’ helpful comments and suggestions. The manuscript has been revised to reflect the changes requested.

Reviewer 1

Ferroptosis is a recently identified type of cell death that is morphologically genetically and mechanistically distinct uniqueness from regulated cell death, including apoptosis, necroptosis, and autophagy.  In this Manuscript, Valeria Consoli et al., investigated mechanism of ferroptosis in different types of breast cancer cells and reveal the importance of heme-oxygenase modulation. Here, author have screened initially basal protein expression , Evaluation of erastin cytotoxicity in breast cancer cells, Ferroptosis model validation, measurement of Ho1 levels after treatment with erastin and Effect of Ho1 silencing on cell viability after treatment with erastin and erastin/hemin combination and then further investigated their effect and different combination studies  such as effect of hemin and erastin on ferroptosis-associated oxidative stress, effect of curcumin-mediated ho1 induction, effect of ferroptosis inducers and ho2 protein expression and effect of ferroptosis inducers on mitochondrial dysfunction and lipid ROS accumulation. In general, well designed experiment, presented well to exhibit ho modulation is a possible novel therapeutic strategy.

Based on the results, minor questions to be addressed.

  1. ROS increase was associated with a significant decrease in GSH cellular content. did you tested that increased ROS prodution cause/result of DNA damage.

We agree with the reviewer comment. Although we didn’t perform the above mentioned analysis yet, it has been demonstrated that labile iron is able to promote tumorigenesis by oxidative stress leading to increased DNA damage and elevated levels of 8-oxo-2’-deoxyguanosine (Seiwert et al.).

  1. Effect of ferroptosis inducers on mitochondrial dysfunction, did you analyzed mitochondrial function (cell mito stress test) using seahorse exptt. This assay provides insight into the cause of mitochondrial dysfunction and an in-depth understanding of metabolic pathways, signals.

We are aware of this type of tests (by seahorse and oroboros instruments) and we would like to perform further metabolic alterations analysis also in 3D cell cultures. To date, we don’t have instrument availability but we sincerely would like to investigate more in depth the role of mitochondria in ferroptosis.

Fig.4a pixel quality need to be improved.

Image resolution has been improved.

original image/blot need to be labeled.

Labels have been added.

Reviewer 2 Report

In this manuscript, the authors have described the role of Heme oxygenation modulation in ferroptosis and its therapeutic potential in triple negative breast cancer. As authors have cited the work, Others have shown that HO overexpression negatively impacts cell proliferation. Ferroptosis has also been demonstrated in breast cancer. It is important to note that the authors simply extended the HO induction approach as an anti-cancer treatment to TNBC. The study provides enough evidence to warrant further studies on HO-induction, especially on naturally occurring compounds. The manuscript can be accepted with revision. 

Major Comments 

  • siRNA experiments in fig 1G- points toward an additional mechanism of action besides HO-1 induction since the viability of HO-1 silenced cells are still significantly impacted with co-treatment of Errastin + Hemin. These results need to be further discussed and commented upon. 
  • The synergistic effect of Hemin and Erastin combination also points toward an additional mechanism of action that needs to be discussed. 
  • A separate discussion section would provide more clarity on the results. In the combined sections, the authors have missed the discussion on certain results, such as Hemin's effect on Ferroptosis biomarker GPX4. 

Minor comments:

  • Fold changes in cell viability curves in the figures do not make sense with the scale presented. The differences seem normalized to control conditions. 
  • Figure 4 A- the resolution is too low to visualize the fluorescence. 
  • A separate discussion section would provide more clarity on the results. 
  • The significant finding the authors emphasize is the cells with low HO-1 expression may benefit from HO induction. HO overexpression has been demonstrated as a mechanism of resistance in cancer as well. Is there a patient population with a lower expression that this approach could help with unmet medical needs?

Author Response

Reviewer 2

In this manuscript, the authors have described the role of Heme oxygenation modulation in ferroptosis and its therapeutic potential in triple negative breast cancer. As authors have cited the work, Others have shown that HO overexpression negatively impacts cell proliferation. Ferroptosis has also been demonstrated in breast cancer. It is important to note that the authors simply extended the HO induction approach as an anti-cancer treatment to TNBC. The study provides enough evidence to warrant further studies on HO-induction, especially on naturally occurring compounds. The manuscript can be accepted with revision.

Major Comments

siRNA experiments in fig 1G- points toward an additional mechanism of action besides HO-1 induction since the viability of HO-1 silenced cells are still significantly impacted with co-treatment of Errastin + Hemin. These results need to be further discussed and commented upon.

In our experimental conditions hemin treatment did not impact on cell viability, nevertheless it is known that hemin is able to induce ROS even at low concentrations (Fiorito et al.). Seiwert and colleagues have shown that HO-1 silencing increased ROS production and enhanced hemin cytotoxicity. These data are in agreement with our results showing that combination treatment of erastin and hemin maintains a cytotoxic effect in HO-1 silenced cells.

The synergistic effect of Hemin and Erastin combination also points toward an additional mechanism of action that needs to be discussed.

Hemin and ferric ammonium citrate can promote erastin-induced ferroptosis, however this effect was not observed by biliverdin or bilirubin treatment (Tang et al.). As reported by NaveenKumar et al., hemin represent a pro-oxidant molecule that can induce ferroptosis both via proteosome and inflammasome activation and Fe2+ accumulation via HO-1 induction. Additionally, low levels of hemin can induce BACH-1 degradation and Nrf2 transcriptional activity in response to augmented oxidative stress levels. Thus, seems evident that hemin contribution in iron release due to heme degradation by HO-1, can synergistically enhance and accelerate erastin cytotoxic effect leading to a further increase in ROS and altering cellular redox homeostasis.

A separate discussion section would provide more clarity on the results. In the combined sections, the authors have missed the discussion on certain results, such as Hemin's effect on Ferroptosis biomarker GPX4.

Hemin treatment caused a significant increase in GPX4 expression levels as also reported by Jin et al.; this result might represent a consequence of the cellular compensatory effect resulting from hemin-induced Nrf2 activation as demonstrated by several studies. Although Nrf2 stabilization occurs following hemin treatment, HO-1 over-activation becomes cytotoxic due to excessive increase of iron beyond ferritin buffering capacity.

Minor comments:

Fold changes in cell viability curves in the figures do not make sense with the scale presented. The differences seem normalized to control conditions.

Figures have been modified and cell viability has been expressed as percentage of control.

Figure 4 A- the resolution is too low to visualize the fluorescence.

Image resolution has been improved.

A separate discussion section would provide more clarity on the results.

Manuscript has been modified as suggested.

The significant finding the authors emphasize is the cells with low HO-1 expression may benefit from HO induction. HO overexpression has been demonstrated as a mechanism of resistance in cancer as well. Is there a patient population with a lower expression that this approach could help with unmet medical needs?

We agree with the reviewer that HO-1 overexpression is generally observed in several human malignant neoplastic diseases, serving as a cytoprotective factor in early stages of tumorigenesis, but eventually promoting malignant cells growth, proliferation, and invasion. However, it has been demonstrated that low levels of HO-1 are associated with an increased risk of metastasis in oral and tongue squamous cell carcinoma together with a significant presence of undifferentiated cells (Yanagawa et al). Additionally, data analysis by Han et al. showed that low expression of HO-1 in renal carcinoma predicted a poor prognosis, which might be improved by activating ferroptosis through induction of HO-1. Furthermore, overexpression of HO-1 in prostate cancer cells resulted in markedly reduced cell proliferation and migration (Gueron et al.). Thus, the influence of HO-1 on cancer proliferation and metastasis is not univocal and may depend on the type of cancer.